# Ingestion of GNSS-Derived ZTD and PWV for Spatial Interpolation of PM_2.5_ Concentration in Central and Southern China

**DOI:** 10.3390/ijerph18157931

**Published:** 2021-07-27

**Authors:** Pengzhi Wei, Shaofeng Xie, Liangke Huang, Lilong Liu

**Affiliations:** 1College of Geomatics and Geoinformation, Guilin University of Technology, Guilin 541006, China; weipengzhi@glut.edu.cn (P.W.); xieshaofeng@glut.edu.cn (S.X.); Lllong99@glut.edu.cn (L.L.); 2Guangxi Key Laboratory of Spatial Information and Geomatics, Guilin 541006, China

**Keywords:** PM_2.5_, PWV, ZTD, GWR, MGWR, empirical Bayesian kriging

## Abstract

With the increasing application of global navigation satellite system (GNSS) technology in the field of meteorology, satellite-derived zenith tropospheric delay (ZTD) and precipitable water vapor (PWV) data have been used to explore the spatial coverage pattern of PM_2.5_ concentrations. In this study, the PM_2.5_ concentration data obtained from 340 PM_2.5_ ground stations in south-central China were used to analyze the variation patterns of PM_2.5_ in south-central China at different time periods, and six PM_2.5_ interpolation models were developed in the region. The spatial and temporal PM_2.5_ variation patterns in central and southern China were analyzed from the perspectives of time series variations and spatial distribution characteristics, and six types of interpolation models were established in central and southern China. (1) Through correlation analysis, and exploratory regression and geographical detector methods, the correlation analysis of PM_2.5_-related variables showed that the GNSS-derived PWV and ZTD were negatively correlated with PM_2.5_, and that their significances and contributions to the spatial analysis were good. (2) Three types of suitable variable combinations were selected for modeling through a collinearity diagnosis, and six types of models (geographically weighted regression (GWR), geographically weighted regression kriging (GWRK), geographically weighted regression—empirical bayesian kriging (GWR-EBK), multiscale geographically weighted regression (MGWR), multiscale geographically weighted regression kriging (MGWRK), and multiscale geographically weighted regression—empirical bayesian kriging (MGWR-EBK)) were constructed. The overall *R*^2^ of the GWR-EBK model construction was the best (annual: 0.962, winter: 0.966, spring: 0.926, summer: 0.873, and autumn: 0.908), and the interpolation accuracy of the GWR-EBK model constructed by inputting ZTD was the best overall, with an average RMSE of 3.22 μg/m^3^ recorded, while the GWR-EBK model constructed by inputting PWV had the highest interpolation accuracy in winter, with an RMSE of 4.5 μg/m^3^ recorded; these values were 2.17% and 4.26% higher than the RMSE values of the other two types of models (ZTD and temperature) in winter, respectively. (3) The introduction of the empirical Bayesian kriging method to interpolate the residuals of the models (GWR and MGWR) and to then correct the original interpolation results of the models was the most effective, and the accuracy improvement percentage was better than that of the ordinary kriging method. The average improvement ratios of the GWRK and GWR-EBK models compared with that of the GWR model were 5.04% and 14.74%, respectively, and the average improvement ratios of the MGWRK and MGWR-EBK models compared with that of the MGWR model were 2.79% and 12.66%, respectively. (4) Elevation intervals and provinces were classified, and the influence of the elevation and the spatial distribution of the plane on the accuracy of the PM_2.5_ regional model was discussed. The experiments showed that the accuracy of the constructed regional model decreased as the elevation increased. The accuracies of the models in representing Henan, Hubei and Hunan provinces were lower than those of the models in representing Guangdong and Guangxi provinces.

## 1. Introduction

PM_2.5_ refers to fine particulate matter with a diameter of less than or equal to 2.5 μm existing in the ambient air; this matter has the characteristics of a long suspension time in the air, a long transportation distance, strong activity, and easy absorption of toxic and harmful substances; especially high PM_2.5_ concentrations cause occurrences of hazy weather [1,2,3]. In recent years, the environmental pollution caused by PM_2.5_ has gradually attracted attention. Since 2012, China has built a large number of ground-based PM_2.5_ stations nationwide. However, China is a vast country, and the number of ground-based PM_2.5_ stations is still scarce, and therefore they cannot accurately explain all the temporal and spatial characteristics of PM_2.5_, thus limiting the application of PM_2.5_ data in a variety of practical applications. Therefore, it is necessary to further study how to obtain continuous and accurate regional PM_2.5_ distributions on different temporal and spatial scales and how to use limited station-derived data to conduct high-precision regional PM_2.5_ temporal and spatial interpolations, and these applications have become a research hotspot.

China is a vast country, and many scholars have tried to explore the variation patterns of PM_2.5_ in different regions of China, such as North China [4], the Yangtze River Economic Zone [5], the Pearl River Delta [6], Heilongjiang Province [7], and the Beijing–Tianjin–Hebei region [8,9]. When exploring the changing characteristics of PM_2.5_, many scholars have tried to use different PM_2.5_-related variables to combine analyses to improve the accuracy of PM_2.5_ simulations. The impact of the population density, GDP and other social or economic activities on PM_2.5_ concentrations in Chinese cities with significant spatial heterogeneity was explored by Gu et al. [10]; Tai et al. used 11-year (1998–2008) PM_2.5_ observation records in the United States to explore the correlations between PM_2.5_ and meteorological variables, and the results showed that the daily variations in meteorological variables described by MLR can explain up to 50% of PM_2.5_ changes [11]; Ye et al. used Fairbanks standard air pollutants (NO_2_, SO_2_, CO, O_3_, PM_2.5_ and PM_10_) and meteorological parameter (temperature, wind speed and relative humidity) observations, temporal changes and related analyses, and all pollutants showed obvious seasonal trends under the influence of climatology, topography and human activities [12]. 

With the development of GNSS technology, the global navigation satellite system’s (GNSS) zenith tropospheric delay (ZTD), zenith wet delay (ZWD) and precipitable water vapor (PWV) data are able to reflect certain atmospheric water vapor information and can also reflect changes in meteorological conditions [13,14]. Therefore, some scholars have begun to introduce GNSS data into the PM_2.5_ research field; for example, Wen et al. [15] explored the correlation between PM_2.5_ and ZWD in Baoding, Hebei, China, and found that the correlation coefficient between the daily average PM_2.5_ and ZWD was mainly greater than 0.4 in autumn and winter in this region, while the correlation coefficient between the hourly average PM_2.5_ and ZWD was mainly greater than 0.3. Guo et al. [16] took the GNSS data from Beijing Fangshan Station (BJFS) as an example to analyze the correlations among GNSS-derived PWV and ZTD and PM_2.5_ hourly sequences. The experimental results showed that it is effective to consider using GNSS-derived ZTD to assist in haze monitoring.

Due to the continuous development and improvement of spatial fitting and interpolation models, some scholars have used geographically weighted regression (GWR) [17] to simulate PM_2.5_ in space and have achieved good results. Zhou et al. [18] established a model based on PM_2.5_ data from 283 prefecture-level cities in China combined with the random regressed effects of the population, economy, and technology, and used geographically weighted regression methods to evaluate the impacts of different factors on haze pollution in different regions. Wang et al. [19] used the GWR method to explore the strengths and directions of the relationships among various factors in Chinese cities and PM_2.5_, established a comprehensive explanatory framework consisting of 18 determinants covering natural and social conditions, and determined three major categories of economic factors and urban characteristics. Among all natural variables, elevation has a statistically significant impact on PM_2.5_ in 95.60% of cities, and is negatively correlated with PM_2.5_ in 99.63% of cities. The effect of elevation is gradually weakened from eastern China to western China; Zou et al. [20] compared the simulation effects of two land use regression (LUR) and GWR models on PM_2.5_ concentrations in California. The results showed that both the GWR and LUR models were able to estimate the PM_2.5_ concentrations and map the spatial distribution in the study area. Jiang et al. [21] introduced a variety of auxiliary variables, such as meteorological and geographic factors, into the GWR model to establish a four-season GWR model of PM_2.5_ in the Yangtze River Delta. Hajilooe et al. [22] evaluated the relationship between the meteorological variables (humidity, pressure, temperature, precipitation and wind speed) associated with the PM_2.5_ concentration in Tehran and environmental parameters (normalized vegetation index and surface temperature from MODIS satellite data) using GWR to evaluate the impacts of key parameters on PM_2.5_ concentrations in winter and summer.

Due to the strong spatial and temporal heterogeneity of PM_2.5_, most scholars have begun to study GWR models that are more applicable to PM_2.5_. Yang et al. [23] analyzed regional PM_2.5_ spatial variation relationships in China by developing a modified GWR model using meteorological, topographical and emission factors observed in 2015. Rui et al. [24] analyzed the spatial distribution of PM_2.5_ concentrations in the Pearl River Delta region of China using an enhanced GWR model by introducing geodetector analysis and principal component analysis (PCA) to enhance the GWR model. Zhai et al. [25] developed a best subset regression (BSR)-augmented PCA-GWR modeling approach to estimate PM_2.5_ concentrations by fully considering the contributions of all potential variables simultaneously, and conducted a one-year experiment comparing the performance of PCA-GWR with that of conventional GWR in the Beijing–Tianjin–Hebei region. The results showed that the PCA-GWR model outperformed the conventional GWR model. 

Most of the abovementioned studies dealt with PM_2.5_ and related variables when modeling the experimental data, but did not consider the effect of model residuals. Some scholars introduced the kriging interpolation method to interpolate the GWR residuals for the purpose of correcting the fitted values of the GWR model for the spatial autocorrelation of the residuals after fitting the GWR model, deriving the GWRK model from this; however, this model has been mostly applied to geological field-related research [26,27,28]. On the basis of GWRK, Kumari et al. [29] proposed a stratified, geographically weighted regression residual kriging (s-GWRK) method and applied it to complex-terrain rainfall interpolations with good results; however, few scholars have applied this type of GWRK model to study PM_2.5_ spatial interpolations.

For the kriging interpolation method, empirical Bayesian kriging (EBK) has been proposed based on this method. Empirical Bayesian kriging differs from other kriging methods in that it accounts for the introduced error by estimating the underlying semivariance function, which has the advantage of predicting standard errors more accurately than other kriging methods, and can accurately predict data that are generally unstable in degree [30,31].

Some scholars have improved the GWR method from the spatial-scale perspective, and the model that is most representative of these improvements is the multiscale geographically weighted regression (MGWR) model, which is more flexible than the GWR model. The model allows different processes to work on different spatial scales [32], and the MGWR model has been applied to spatial simulations of PM_2.5_ by some; Fan et al. [33] used the MGWR model to simulate the spatial and temporal patterns of PM_2.5_ and its associated influencing factors during the outbreak of new crown pneumonia in China. Yan et al. [34] simulated the spatial and temporal distribution characteristics and driving forces of PM_2.5_ in three major urban agglomerations in the Yangtze River Economic Zone of China using the MGWR model, and found that the total precipitation, wind speed and green coverage had the most significant effects on the PM_2.5_ distribution.

Although both types of models, GWR and MGWR, have performed well in previous spatial studies of PM_2.5_, as described above, strong spatial heterogeneity and spatial nonstationarity exists in PM_2.5_ distributions, and it is difficult for the above-mentioned models to handle or simultaneously handle these two PM_2.5_ distribution characteristics. Therefore, to improve the interpolation of PM_2.5_ concentration values, we compare the changes induced by the differences in each time scale and spatial scale to the model interpolation effect, fully consider the variations in GNSS-derived ZTD and PWV, and two models, the kriging and empirical Bayesian kriging models, are introduced to eliminate the effect of residual spatial correlations on the fitting of the GWR and MGWR models, thus improving the interpolation accuracy.

## 2. Materials and Methods

### 2.1. Methodology

#### 2.1.1. Geographically Weighted Regression (GWR) Model

Geographically weighted regression (GWR) is a spatial analysis technique that belongs to the local spatial analysis model and is a relatively simple and effective method used to detect spatial nonsmoothness. The method allows for the existence of different spatial relationships in different geographic spaces and results in local, rather than global, parameter estimates, thus enabling the detection of the spatial nonsmoothness of spatial data.

GWR explores the spatial variability in the study object at a given scale and the associated drivers by establishing local regression equations at each point in the spatial scale. As this method takes into account the local effects of spatial objects, it has the advantage of improved accuracy, and its basic principle can be expressed as follows [17,18,19,20,21,22]:$$Y_{i}=\beta_{0}(u_{i},v_{i})+\sum_{k=1}^{p}\beta_{k}(u_{i},v_{i})x_{ik}+\varepsilon_{i}$$
where $\beta_{0}(u_{i},v_{i})$ is the regression constant of the model at position $\left(u_{i},v_{i}\right)$ (intercept term); $\left(u_{i},v_{i}\right)$ is the coordinate of the ith sampling point; $\beta_{k}(u_{i},v_{i})$ is the kth regression parameter at the ith sampling point; $x_{ik}$ is the kth influence variable for the ith observation; p is the number of influencing variables; $\varepsilon_{i}$ are the regression residuals.

The estimation of the regression coefficients in the model is implemented using the least squares method, and the coefficients at each point are represented by the matrix as follows:$$\beta (u_{i},v_{i})={\left[X^{T}W(u_{i},v_{i})X\right]}^{-1}X^{T}W(u_{i},v_{i})Y$$
where $W(u_{i},v_{i})$ is the diagonal matrix of spatial weights, X is the design matrix of the independent variable, and Y is the vector of dependent variables.

The spatial weight matrix W is estimated using a Gaussian function:$$\left\{ \begin{array}{l} W_{ij}={\left[1-{\left(d_{ij}/h\right)}^{2}\right]}^{2},d_{ij}<h\\ W_{ij}=0,d_{ij}\geq h \end{array}\right.$$
where $W_{ij}$ is the weight of the spatially known point j when estimating the point i to be measured; $d_{ij}$ is the Euclidean distance between point i to be estimated and sample point j; h is the bandwidth, which is judged using the AICc (corrected Akaike Information Criterion).

#### 2.1.2. Geographically Weighted Regression Kriging (GWRK) Model

The geographically weighted regression kriging (GWRK) model uses the kriging model to spatially interpolate the regression residuals obtained from the GWR model; then, the obtained residual interpolation results are superimposed on the GWR regression estimates to correct the fitter GWR-derived values to obtain the GWRK model estimation results [26,27,28,29]:$$GWRK_{pm2.5}=GWR_{pm2.5}+K{\mathrm{riging}}_{\mathrm{GWR}\;\mathrm{RES}}$$
where $GWRK_{\mathrm{pm}2.5}$ is the PM_2.5_ concentration estimated by the GWRK model; $GWR_{\mathrm{pm}2.5}$ is the regional PM_2.5_ concentration estimated by the GWR model; $K{\mathrm{riging}}_{\mathrm{GWR}\;\mathrm{RES}}$ is the regional residual result obtained by kriging interpolation of the regression residuals after the PM_2.5_ concentration values are estimated by the GWR model.

#### 2.1.3. Geographically Weighted Regression—Empirical Bayesian Kriging (GWR-EBK) Model

The empirical Bayesian kriging (EBK) method is a geostatistical interpolation method that automatically performs the most difficult steps in the construction of an effective kriging model. EBK automatically calculates the model parameters by constructing subsets and simulating the process.

Empirical Bayesian kriging differs from other kriging methods in that it accounts for the error introduced by estimating the underlying semivariance function. Other kriging methods calculate a semivariance function from known data locations and use this single semivariance function to make predictions at unknown locations; this process implicitly assumes that the estimated semivariance function is the true semivariance function in the interpolated region. By not accounting for the uncertainty in the estimation of the semivariance function, other kriging methods underestimate the standard error of the prediction.

*EBK* predicts standard errors more accurately than other kriging methods for data that are generally unstable, especially for small data sets [30,31], and is calculated as follows:$$GWR-EBK_{\mathrm{pm}2.5}=GWR_{\mathrm{pm}2.5}+EBK_{\mathrm{GWR}\;\mathrm{RES}}$$
where $GWR-EBK_{\mathrm{pm}2.5}$ is the *GWR*-*EBK* model estimate of the PM_2.5_ concentration, $GWR_{\mathrm{pm}2.5}$ is the regional PM_2.5_ concentration estimated by the *GWR* model, and $K{\mathrm{riging}}_{\mathrm{GWR}\;\mathrm{RES}}$ is the regional residual result obtained by *EBK* interpolation of the regression residuals from the *GWR* model after estimating the PM_2.5_ concentration values.

#### 2.1.4. Multiscale Geographically Weighted Regression (MGWR) Model

The specific bandwidth of each variable in the MGWR model can be used as an indicator of the spatial scale of each spatial processing action. The MGWR model is calculated as follows [32,33,34]:$$Y_{i}=\sum_{k=1}^{p}\beta_{b\omega k}(u_{i},v_{i})x_{ik}+\varepsilon_{i}$$
where $\left(u_{i},v_{i}\right)$ is the coordinate of the ith sampling point; $\beta_{k}(u_{i},v_{i})$ is the kth regression parameter at the ith sampling point; each regression coefficient $\beta_{b\omega k}$ is the bandwidth of variable k based on a local regression with a specific bandwidth that represents the difference between the MGWR and GWR results. MGWR models allow optimizing specific bandwidths for the relationships between different independent and dependent variables.$x_{ik}$ is the kth influencing variable for the ith observation; p is the number of influencing variables; $\varepsilon_{i}$ are the regression residuals.

MGWR uses the same Gaussian kernel function as the previous GWR. We mainly use the MGWR2.2 software (The School of Geographical Sciences and Urban Planning at Arizona State University, Tempe, AZ, USA).

#### 2.1.5. Multiscale Geographically Weighted Regression Kriging (MGWRK) Model

The *MGWRK* model uses the kriging model to spatially interpolate the regression residuals obtained from the *MGWR* model; then, the obtained residuals are interpolated to correct the *MGWR* estimates:$$MGWRK_{\mathrm{pm}2.5}=MGWR_{\mathrm{pm}2.5}+K{\mathrm{riging}}_{\mathrm{MGWR}\;\mathrm{RES}}$$
where $MGWRK_{\mathrm{pm}2.5}$ is the *MGWRK* model estimate of the PM_2.5_ concentration, $MGWR_{\mathrm{pm}2.5}$ is the regional PM_2.5_ concentration estimated by the *MGWR* model, and $K{\mathrm{riging}}_{\mathrm{MGWR}\;\mathrm{RES}}$ is the regional residual result obtained via kriging interpolation of the regression residuals from the *MGWR* model after estimating the PM_2.5_ concentration values.

#### 2.1.6. Multiscale Geographically Weighted Regression—Empirical Bayesian Kriging (MGWR-EBK) Model

The MGWR-EBK model uses an empirical Bayesian kriging model to spatially interpolate the regression residuals obtained from the *MGWR* model; then, the obtained residuals are interpolated to correct the *MGWR* estimates:$$MGWR-EBK_{\mathrm{pm}2.5}=MGWR_{\mathrm{pm}2.5}+EBK_{\mathrm{MGWR}\;\mathrm{RES}}$$
where $MGWR-EBK_{\mathrm{pm}2.5}$ is the *MGWR*-*EBK* model estimate of the PM_2.5_ concentration, $MGWR_{\mathrm{pm}2.5}$ is the regional PM_2.5_ concentration estimated by the *MGWR* model, and $EBK_{\mathrm{MGWR}\;\mathrm{RES}}$ is the regional residual result obtained by *EBK* interpolation of the regression residuals from the *MGWR* model after the PM_2.5_ concentration values are estimated.

### 2.2. Study Area and Data

#### 2.2.1. Study Area

The five provinces of south-central China (Henan, Hubei, Hunan, Guangdong and Guangxi) are located within longitudes of 104°28′–117°19′ E and latitudes of 20°13′–36°22′ N. These provinces cover an area of nearly one million square kilometers and have a resident population of nearly 400 million people; the terrain is high in the west and low in the east. Due to the large north–south span and the very different folklore and geographical styles of each province, the region is an important economic and cultural development area in China; thus, the monitoring and prevention of environmental pollution and other events should be the focus in the process of development. 

The data obtained from ground monitoring stations have the advantage of high accuracy, so this paper mainly uses various types of station data for experiments and analysis. The missing data from the observation stations in the five south-central provinces were screened and eliminated. Finally, annual average air quality parameter data (PM_2.5_, CO, SO_2_, NO_2_, and O_3_) collected at PM_2.5_ ground stations in five provinces in south-central China from 2017 to 2019 and quarterly average air quality parameter values from December 2016 to November 2019 (Figure 1a) were used for five provinces in south-central China (data from http://envi.ckcest.cn/environment/; accessed on 9 April 2021).

GNSS (global navigation satellite system)-derived data (PWV and ZTD) were collected from 16 sounding stations and 21 GNSS sites in south-central China for annual average values from 2017 to 2019 and quarterly average values from December 2016 to November 2019, respectively (Figure 1b). (PWV from http://weather.uwyo.edu/upperair/sounding.html, accessed on 31 March 2021; ZTD from http://www.cgps.ac.cn/, accessed on 31 March 2021).

Annual averages from 2017 to 2019 and seasonal averages from December 2016 to November 2019 for all types of meteorological data (barometric pressure, temperature and wind speed) for South Central China from 97 meteorological stations (Figure 1c) in the region (data from http://data.sheshiyuanyi.com/WeatherData/, accessed on 13 April 2021), and the digital elevation model (DEM)-derived (with a spatial resolution of 250 m) elevation values of the region were used as the experimental data. To simplify the expression, all meteorological parameters used in this paper are abbreviated: PRE stands for pressure, TEM stands for temperature, and WIN stands for wind speed. Table 1 summarizes the sources and timestamps of all variables.

#### 2.2.2. Data Preprocessing

ZTD refers to zenith tropospheric delay; for space geodesy, this delay causes signal propagation errors, which affect the observation accuracy and are a kind of error source. In 1992, Bevis et al. [35] proposed the precipitable water vapor (*PWV*) method using a ground-based GPS (global positioning system). *PWV* is mainly calculated from the *ZWD* as follows [13,14]:PWV=Π⋅ZWD
where Π is a conversion factor.

The number of GNSS sites, sounding stations, and meteorological stations is much lower than the number of PM_2.5_ ground stations. To better obtain data on *ZTD*, *PWV,* and meteorological parameters (temperature, barometric pressure and wind speed) with a corresponding time scale to that obtained at the PM_2.5_ ground stations, three interpolation methods, IDW (inverse distance weighting), OK (ordinary kriging) and TSF (tension spline function), were used, and their interpolation effects were compared when interpolating different variables at annual and seasonal scales in the study region.

Inverse distance weighting (IDW) is an interpolation method that weights the distance between the interpolation point and the sampling point. The method is simple and effective; the closer the interpolation point is, the larger the weight is, and the contribution of the weight is inversely proportional to the distance [10]. The calculation formula is as follows:$$Y=\frac{\sum_{i=1}^{n}\frac{1}{\left(D_{i}\cdot p\right)}\cdot Y_{i}}{\sum \frac{1}{D_{i}}\cdot p}$$
where Y is the estimated value of the interpolation point; $Y_{i}$ indicates the measurement sample value; n is the number of measurement samples involved in the calculation; $D_{i}$ denotes the distance between the interpolation point and station i; p is the weight of the distance.

The ordinary kriging model is expressed as the following equation [36].
$$Z_{v}^{*}(x)=\sum_{i=1}^{n}\lambda_{i}Z(x_{i})$$
where $x_{i}$ is the position of any point in the study area and $\lambda_{i}$ is the weighting factor, which denotes the contribution of each known sample value $Z(x_{i})$ to the kriging estimate $Z_{v}^{*}(x)$.

The tension spline function (TSF) is a kind of radial basis function interpolation method that is fast, and its range of estimated sizes is not limited. The basic principle of the tension spline function is shown in the following equation:$$\left\{ \begin{array}{l} S(x,y)=a+\sum_{j=1}^{N}\lambda_{j}R(r_{j}),j=1,2....,N\\ R(r)=-\frac{1}{2\pi \varphi^{2}}[\ln(\frac{r\varphi }{2})+c+k_{0}(r\varphi )] \end{array}\right.$$
where S(x,y) is the interpolation result; a is the trend function; N is the number of points in the interpolation area; $\lambda_{j}$ is the coefficient obtained by solving the system of linear equations; $r_{j}$ is the distance from point (x,y) to the jth point; φ is the weight parameter; $k_{0}$ is the modified Bessel function; c is a constant (c≈0.577215).

The *RMSE* (root mean square error) values of the data observed at the corresponding stations were calculated for the accuracy assessment from three models:$$RMSE=\sqrt{\frac{1}{n}\sum_{i=1}^{n}{\left(X_{i}-{\hat{X}}_{i}\right)}^{2}}$$
where n is the sample size, $X_{i}$ is the actual observed value, and $\hat{X_{i}}$ is the model interpolation result. The *RMSE* results are shown in Figure 2.

By looking at the information in Figure 2, we can see that the RMSE values of the three interpolation methods are low because after the regional interpolation of the site data, when the interpolation results are extracted to the original site, the data changes are small, so the differences between the interpolated results and the original site data are also small; however, according to the data in Figure 2, we can see that all three interpolation methods can be applied to the regional interpolation of the variables in the table. However, the best effect is the inverse distance weighted interpolation method, followed by the tensor spline function interpolation method; the ordinary kriging method has a relatively poor effect. Therefore, to better obtain meteorological data from PM_2.5_ ground stations and GNSS-derived PWV and ZTD data, the inverse distance weighted interpolation method is uniformly used for regional interpolation processing.

## 3. Spatial and Temporal Characteristics of PM_2.5_ in South-Central China

### 3.1. Spatial and Temporal Distributions of PM_2.5_

Investigating the spatial and temporal distributions of PM_2.5_ is a very important part of the experiments conducted in this study. To better explore the variation in regional PM_2.5_ values, the PM_2.5_ data obtained from each air quality monitoring station were interpolated with the inverse distance weighting method to obtain a regional PM_2.5_ distribution map, which was analyzed at two temporal resolutions: annual and seasonal.

As shown in Figure 3a, there is an obvious high distribution pattern in the north and a low pattern in the south; the more serious pollution is located in Henan and Hubei provinces, and the lighter pollution is located in Hunan, Guangxi and Guangdong provinces, which is consistent with the conclusion of the previous analysis. From the annual average values, the annual average PM_2.5_ values in the central and southern regions range from roughly 20 μg/m^3^ to 80 μg/m^3^, with a relatively smooth change.

As we can see from Figure 3b, pollution in winter is the most serious, followed by that in spring (Figure 3c) and autumn (Figure 3e), while the lightest pollution occurs in summer (Figure 3d); winter pollution is mainly concentrated in Henan and Hubei provinces. The five provinces in south-central China have better air quality in summer (Figure 3d) as long as the PM_2.5_ values vary below 60 μg/m^3^.

After exploring the spatial distribution pattern of PM_2.5_ in the south-central region, a cluster analysis of PM_2.5_ values in the region was conducted to obtain the spatial clustering information of elements with high or low values to achieve a deeper understanding of PM_2.5_ distribution characteristics in the south-central region.

### 3.2. PM_2.5_ Clustering Analysis

Cluster analysis reveals how high or low variables are clustered in a region and is one of the most important tools used to explore spatial patterns. We used the anselin local moran’s I statistic in Arcgis 10.4 to perform a clustering analysis of PM_2.5_ in the five study provinces in south-central China, where the spatial relationship of elements was defined by choosing the inverse distance, and the distance between each element and the adjacent elements was calculated by choosing the euclidean distance; the results are shown in Figure 4.

As shown in Figure 4a, high clustering of PM_2.5_ in the south-central region mainly occurs in the Henan and Hubei regions, while low clustering is mainly concentrated in south-central Guangdong province, and there is almost no clustering in the Guangxi or Hunan regions at the annual average scale.

Additionally, as shown in Figure 4, the seasonal average scale clustering results are similar to the annual average-scale clustering results, showing obvious high clustering in Henan and low clustering in Guangdong, while in winter (Figure 4b), local areas in Hubei also show high clustering, and in spring (Figure 4c), local areas in Hunan and Guangxi provinces also show low clustering, while in autumn (Figure 4e), low clustering occurs in the northwestern part of Hunan, while high clustering was observed in the eastern part.

## 4. Variable Selection

After exploring the spatial and temporal variation patterns of PM_2.5_ in the five studied south-central provinces at different scales, a correlation analysis of the multiple variables considered in the experiment was also conducted, before the modeling analysis was performed, to ensure that all the variables considered had a certain correlation with PM_2.5_ and were thus suitable for subsequent modeling work.

### 4.1. Analysis of the Relationships between the Original-Sequence ZTD and PWV with PM_2.5_

Satellite-derived tropospheric data have a strong correlation with PM_2.5_ [37]. To explore the variation patterns presented by PM_2.5_ with PWV and ZTD at different time scales at each station in the five provinces of south-central China, data corresponding to 340 PM_2.5_ ground stations were mapped and analyzed.

#### 4.1.1. Annual Average Scale

As seen in Figure 5, the variation range of the ZTD values in the five provinces of south-central China is generally between 2300 and 2600 mm, while the PWV values mainly vary between 20 and 50 mm, and the overall trend of PWV is clearer than that of ZTD, while the variation patterns of ZTD and PWV both show obvious negative correlations with PM_2.5_.

#### 4.1.2. Seasonal Average Scale

From Figure 6, it seems that ZTD is the lowest in winter, and the variation range is small, basically varying from approximately 2300–2500 mm; in summer, the value is the highest, with a main variation range between 2400 and 2700 mm, while in spring and autumn, ZTD basically stays between 2300 and 2600 mm, and the overall variation trends of PM_2.5_ and ZTD at the seasonal average scale are consistent with those at the annual mean scale. The overall trends of PM_2.5_ and ZTD at the seasonal mean scale are consistent with the negative-correlation phenomenon observed at the annual mean scale.

As seen in Figure 7, PWV and ZTD seem to be the lowest and have the smallest variation ranges in winter, with variation ranges below 30 mm; the highest values and largest variation ranges are observed in summer, with values varying above 40 mm, while the variations in PWV and PM_2.5_ recorded in spring and autumn range between 20 mm and 45 mm, and the variation pattern between PWV and PM_2.5_ is consistent with the negative correlation phenomenon observed under the annual average scale.

### 4.2. Spearman Correlation Analysis of Each Variable with PM_2.5_

Since Spearman’s correlation coefficient determination rule does not require a given distribution of original variables and has a wide range of applications [38], this method is applied here for the correlation analysis; the annual average data for three years (from 2017 to 2019) for each type of variable and all quarterly average data from December 2016 to November 2019 for a total of 15 time stamps were used for correlation analysis; the results are shown in Figure 8.

Because of the large number of explanatory variables considered in the experiments, this paper presents the various types of correlation analysis in a uniform way using line graphs, so that the five time scales of annual average, winter, spring, summer and autumn can be clearly visualized.

The results shown in Figure 8 reveal that the two types of factors, ZTD and PWV, show high rankings and negative correlations with PM_2.5_, reflecting the previous conclusion.

On the annual average scale, the correlation level of the wind speed is the lowest, the correlation levels of the four air pollutants are maintained at approximately 0.5, and the correlation level of the DEM is between 0.3 and 0.4. Among the meteorological factors, the correlation levels of the wind speed and air pressure are low, while the correlation level of temperature is high and stable, with values maintained above 0.8.

On a seasonal scale, temperature, ZTD and PWV still maintain high negative correlations in winter; in spring, the correlation levels of all air pollutants are roughly the same as the annual average values; in summer, the correlation level of O_3_ is higher than those of the other pollutants, and the correlation level of the DEM is the highest compared with the annual average and with other seasons; in autumn, the correlation level of O_3_ drops sharply compared with that measured in summer.

### 4.3. Geodetectors

Stratified heterogeneity is one of the basic characteristics of geographical phenomena and the spatial expression of natural and socioeconomic processes, and refers to the fact that the value of an attribute varies among different regions. PM_2.5_, which is a class of variables with strong spatial heterogeneity, also has stratified heterogeneity, and the q statistic of a geodetector can be used to measure this stratified heterogeneity, detect its explanatory factors, and analyze the interactive relationships among variables. The *q* value is calculated as follows [39]:$$q=1-\frac{1}{N\sigma^{2}}\sum_{h=1}^{L}N_{h}\sigma_{h}^{2}$$

The region is divided into *h* = 1, …, *L* layers, i.e., *L* subregions; *N* and $\sigma^{2}$ denote the overall size and its variance, respectively; the *q* value has a clear physical meaning: the magnitude of the *q*-value indicates the percentage of variance in attribute y that is explained by stratification x.

Therefore, in an effort to investigate the stratified heterogeneity of PM_2.5_ and to detect the extent to which a given factor explains the stratified heterogeneity of PM_2.5_, geographic probes were used to detect the q-values of each variable; the results are shown in Figure 9a.

Figure 9a shows that meteorological factors and GNSS-derived ZTD and PWV show strong abilities to explain the stratified heterogeneity of PM_2.5_, while atmospheric pollutants and the DEM have the second-strongest abilities to explain the stratified heterogeneity of PM_2.5_. Among them, SO_2_, which has a strong correlation with PM_2.5_ as well as high exploratory regression significance, has a weaker ability to explain the stratified heterogeneity of PM_2.5_.

### 4.4. Exploratory Returns

To further investigate whether the effect of each variable on PM_2.5_ is significant, we then conducted an exploratory regression experiment, in which all possible combinations of the input candidate explanatory variables were evaluated to obtain the ratio of the number of times each alternative explanatory variable was statistically significant to its statistical significance, where the explanatory variables with stronger effects were always significant, which could help to find the variables that were always strong explanatory factors. The results of the exploratory regression for the significance of each variable are shown in Figure 9b.

From Figure 9b, it can be seen that the significance magnitudes can be ranked as atmospheric pollutants > GNSS parameters (ZTD and PWV) > meteorological factors, thus further illustrating the importance of GNSS-derived PWV and ZTD in reflecting PM_2.5_ changes; the significance of PWV on PM_2.5_ is 100% in winter time, and that of ZTD on PM_2.5_ in summer time also almost remained at 100%, so these two types of factors showed good correlations with PM_2.5_ changes in the five studied south-central provinces.

### 4.5. Multicollinearity Test

It is important to test for multicollinearity in the considered variables before constructing a geographically weighted class model. The combination of variables that are suitable for modeling is selected according to the diagnostic results, facilitating the smooth expression of the model. Therefore, here, a multicollinearity test was performed for all the variables preconsidered in this paper, and the results are shown in Figure 10.

We generally consider a VIF (variance inflation factor) value less than 10 to indicate a small amount of multicollinearity, which does not affect the modeling results. Therefore, Figure 10 shows that the VIF values between O_3_, CO, NO_2_, SO_2_, DEM, wind, and the barometric pressure are not sufficient to trigger multicollinearity. Different degrees of multicollinearity problems exist among the temperature, PWV, and ZTD at the annual scale and at seasonal scales other than summer, especially in winter. The reason for this result may be that the temperature, PWV and ZTD in the five south-central provinces maintain high correlations with PM_2.5_ on both the annual and seasonal scales, so these three types of variables also show strong correlations with each other, and the subsequent modeling experiments must model and discuss these three types of variables separately. The modeling variable combinations can then be divided into three separate categories, which are applied separately for the subsequent model construction; their variable combinations are shown in Table 2.

## 5. Results and Analysis

### 5.1. Overall Model Effect

Six types of models (GWR, GWRK, GWR-EBK, MGWR, MGWRK and MGWR-EBK) were analyzed on annual and seasonal scales for each of the three combinations of variables. To quantitatively compare the overall simulation effect of each model at each scale, the decidability factor *R*^2^ is used as the basis for comparison, and is calculated as shown in Equation:$$R^{2}=1-\frac{\sum_{i}{\left(\hat{y_{i}}-y_{i}\right)}^{2}}{\sum_{i}{\left(\bar{y_{i}}-y_{i}\right)}^{2}}$$
where $y_{i}$ is the real value, $\hat{y_{i}}$ is the interpolated result, and $\bar{y_{i}}$ is the mean of the real values. The value of *R*^2^ is between 0 and 1, and the larger the value is, the better the model fitting effect is. The *R*^2^ value of each model for each scale is listed in Figure 11.

The information in Figure 11 shows that on an annual average scale, all six models can reach *R*^2^ values above 0.9 using the three variable schemes for modeling, and show good fitting effects. Among the six models analyzed using variables such as temperature, the lowest *R*^2^ values are obtained for the two models MGWR and MGWRK, with values of 0.922 recorded, and the highest value is obtained for the GWR-EBK model, with an *R*^2^ value of 0.962 recorded. The lowest *R*^2^ when modeling with variables such as PWV is obtained for MGWR at 0.929. This lowest value is slightly higher than the lowest *R*^2^ value of the model obtained when considering variables such as temperature, while the highest *R*^2^ value is 0.962 for the GWR-EBK model, which is the same as the highest value obtained when considering variables such as temperature. When modeling with variables such as ZTD, the lowest value is 0.914 for the MGWR model, and the highest value is still 0.962 for the GWR-EBK model; therefore, on the annual average scale, the best overall model fit is found for the GWR-EBK model, and the worst is the MGWR model.

From the analysis of the seasonal average scale, the GWR-EBK model effect reaches 0.964 in winter when the six models consider variables such as temperature due to the annual average scale of 0.962, while the fitting effect of MGWR drops to 0.891. When considering variables such as PWV, the fit of the GWR-EBK model improves again compared to the combination of temperature variables, reaching 0.966, while the MGWR model decreases again to 0.879. When considering variables such as ZTD, the *R*^2^ values of the MGWR-class models all drop below 0.9, while the GWR-EBK model fit is the same as that obtained when considering PWV-like variables.

The *R*^2^ values of all six models decrease in spring compared to winter, and the GWR-EBK model still performs best in spring among all six models. The effect of the combination of both variables considering temperature and ZTD outperformed the effect of the combination of variables considering PWV, reaching 0.928. In contrast, the MGWR model with a combination of temperature-class variables had the worst simulation fit, with an *R*^2^ of 0.838. The *R*^2^ values of 0.846 and 0.856 obtained when considering the combination of two types of variables, PWV and ZTD, respectively, are better than the modeling effect obtained via the combination of temperature variables.

In summer, when PM_2.5_ values change to the lowest values measured among the four seasons, the *R*^2^ values of the six models continue to fall, all to below 0.9, with the highest *R*^2^ value of 0.874 obtained for the GWR-EBK model when considering the combination of ZTD variables, and the lowest *R*^2^ value of 0.754 obtained for the MGWR model.

In autumn, appropriate increases in *R*^2^ are observed for the six models. Among them, the GWR-EBK model has an *R*^2^ value higher than 0.9 for all three variable combinations modeled, with the highest value being modeled using the ZTD variable combination (0.910), and the lowest *R*^2^ value of 0.792 obtained for the MGWR model built using the PWV variable combination.

In summary, in terms of the overall model goodness of fit, the model effects are annual > winter > spring > autumn > summer, the best performance of the goodness of fit is obtained with the GWR-EBK model, and the worst performance is obtained with the MGWR model. Compared with the results obtained with single models such as GWR and MGWR, the quadratic treatment of single-model-regression residuals using interpolation methods such as kriging and empirical Bayesian kriging can effectively improve the fit of the original data. From the choice of variable combinations, the highest *R*^2^ value is 0.962 for all three variable combinations at the annual average scale, and the effect of considering the combination of ZTD variables is better than the effects of considering the other two types of variable combinations, temperature and PWV, at the seasonal average scale. The possible reason for the change in the model *R*^2^ values with the changing seasons is that when the PM_2.5_ value is lower in a given season, the decrease ratio of the sum of squares of the difference between the true value and the mean value may be greater than the decrease ratio of the sum of squares of the residuals, so the increase in the sum of squares of the residuals divided by the sum of squares of the difference between the true value and the mean value leads to a decrease in the *R*^2^ value.

We calculated and summarized the RMSEs of the interpolation results obtained from the various constructed models; RMSE is a good indicator for testing the accuracy of an interpolation and can be used to measure the deviation between the interpolation results and the true values. The RMSE results are shown in Figure 12.

As seen from the results in Figure 12, the GWR-EBK model with the best goodness of interpolation on the annual average scale also has the smallest RMSE, but unlike the goodness of interpolation results, the GWR-EBK model has an equal goodness of interpolation after considering three different combinations of variables for modeling, while in the RMSE results, the combination of ZTD variables considered for modeling results in a smaller RMSE with better results.

On the seasonal average scale, the RMSEs of the models also show a larger phenomenon in winter due to the larger PM_2.5_ values, with the lowest value obtained for the GWR-EBK model built with the combination of PWV variables (an RMSE of 4.5 μg/m^3^) and the highest obtained for the MGWR model built with the combination of ZTD variables (an RMSE of 10.4 μg/m^3^). The RMSE of the GWR-EBK model built by the combination of ZTD variables and temperature variables was optimal and equal in spring and autumn, with RMSEs of 3.2 μg/m^3^ and 3.1 μg/m^3^ obtained, respectively; the PM_2.5_ values were the lowest in summer, so the RMSEs of the model interpolation were also the lowest in this season; the RMSE of the GWR-EBK model obtained using the combination of three types of variables was the lowest and was equal to 2.7 μg/m^3^.

The model effects used for the comparison of the RMSE results are quite similar to those used for the comparison of the goodness of interpolation results. The GWR-EBK model performs the best, and at the annual average scale, the combined results of the two comparisons show that the GWR-EBK model constructed by considering the combination of ZTD variables works best. At the seasonal average scale, the GWR-EBK model constructed by considering the combination of PWV variables performed best in winter, while the GWR-EBK model constructed by the combination of ZTD variables had the highest accuracy in spring, summer and autumn.

### 5.2. Local Model Effect

PM_2.5_ is a variable with strong spatial and temporal heterogeneity. To further analyze the spatial effect of the model interpolation for PM_2.5_ in depth, the model effect is further explored from two perspectives: the DEM-derived elevation and the province.

#### 5.2.1. Elevation as a Classification Criterion

The 340 PM_2.5_ ground stations in the five south-central provinces of China were roughly divided into three categories based on their DEM-derived elevations and the number of stations, with 117 stations below 35 m, 111 stations between 35 and 90 m, and 112 stations greater than 90 m.

The accuracy of the model interpolation for each local elevation interval was quantified using RMSE, and the obtained precisions were compared.

From the annual average data, it can be seen that with an increase in the DEM elevation, the overall accuracy of the model interpolation gradually decreases; for the two elevation intervals less than 35 m (Figure 13a–c) and more than 90 m (Figure 13g–i), the GWR-EBK model built from the combination of the three variables has the highest interpolation accuracies, at 2.3 μg/m^3^ and 2.7 μg/m^3^, respectively. In the interval of 35–90 m (Figure 13d–f), the accuracy of the GWR-EBK model built by the combination of two variables, PWV and ZTD, is better than that of the model constructed by the combination of temperature variables.

The RMSE statistics of the model considered for each elevation interval in winter show that the accuracy of the GWR-EBK model established by using the combination of PWV variables is optimal in each elevation interval, showing good generalizability, and combined with the overall accuracy evaluation results above, this result proves that the accuracy of the model is optimal in winter in the five provinces in south-central China, both overall and in different elevation intervals.

From the statistical results obtained in spring, it can be seen that the GWR-EBK model built considering the combination of two types of variables, temperature and ZTD, has the highest interpolation accuracy, with RMSEs of 2.4 μg/m^3^ and 3.2 μg/m^3^ recorded in the intervals less than 35 m (Figure 13a,c) and 35–90 m (Figure 13d,f), respectively, while in the interval greater than 90 m (Figure 13g,i), the GWR-EBK model built by considering the combination of temperature variables has the highest interpolation accuracy (a value of 3.4 μg/m^3^). From the elevation interval effect, it can be seen that although the GWR-EBK model established by considering the combination of temperature variables and ZTD variables in the overall accuracy evaluation of the model in the previous section has an equal RMSE value, after refining the elevation interval, it can be found that the effect of considering the combination of temperature variables (Figure 13g) is better than the effect of considering the combination of ZTD variables (Figure 13i) in the interval greater than 90 m.

The model interpolation accuracy results presented in the summer were consistent with those obtained on the annual average scale, and the GWR-EBK model built by the three variable combinations had the highest accuracies of 2.1 μg/m^3^ and 2.8 μg/m^3^ in the two elevation intervals less than 35 m (Figure 13a–c) and more than 90 m (Figure 13g–i), respectively. In the 35–90-m interval, the GWR-EBK model built by the combinations of the PWV (Figure 13e) and ZTD (Figure 13f) variables had better accuracy than that of the temperature variable (Figure 13d) combination constructed models. The accuracy of the model interpolation remained consistent overall, and for each elevation interval during the fall season: i.e., the GWR-EBK model consisting of a combination of two types of variables, temperature and ZTD, was considered to be the best.

#### 5.2.2. Province as a Classification Criterion

The 340 PM_2.5_ ground stations were then divided into provinces, and the model effects were discussed in each of the five provinces in the south-central region to explore the model applicability of each province at each scale, including 66 points in Henan province, 48 points in Hubei province, 75 points in Hunan province, 101 points in Guangdong province, and 50 points in Guangxi province. The RMSE results obtained for the six models constructed from the combination of the three variable types at the annual average scale are listed in Figure 14.

Through Figure 14, we can see that the model has larger RMSE values in Henan, Hubei and Hunan provinces and smaller RMSE values in Guangdong and Guangxi provinces, which is consistent with the previously obtained PM_2.5_ distribution patterns in the five south-central provinces (high in the north and low in the south).

The best results were obtained for the GWR-EBK model constructed by combining two types of variables, PWV and ZTD, in Henan province (Figure 14a); for the GWR-EBK model constructed by combining three types of variables in Hubei province (Figure 14b); for the GWR-EBK model constructed by combining two types of variables, temperature and ZTD, in Hunan province (Figure 14c); for the model constructed by combining temperature variables in Guangdong province (Figure 14d); for the model constructed by combining PWV variables in Guangxi province (Figure 14e).

In winter, the GWR-EBK model constructed from the combination of PWV variables had the best applicability in all five provinces, and combined with the results of the previous analysis, this proves that the model has the strongest generalizability during winter in the five south-central provinces.

In two provinces, Hubei (Figure 14b) and Guangdong (Figure 14d), in spring, the GWR-EBK models constructed with different combinations of variables were equally applicable; in Henan province (Figure 14a), the consideration of the combination of both temperature and ZTD variables obtained better results; in Hunan province (Figure 14c), the best GWR-EBK model was constructed by considering ZTD; in Guangxi province (Figure 14e), the best GWR-EBK model was constructed by considering the combination of temperature variables.

The models with the best interpolation accuracies for Henan province (Figure 14a) in summer are of four types: the GWR-EBK and MGWR-EBK models considering PWV, the MGWR-EBK model considering temperature and the GWR-EBK model considering ZTD; combined with the *R*^2^ values of the summer models, these results show that the GWR-EBK model considering ZTD is the best overall. The GWR-EBK model constructed by the combination of three variables in Hubei province (Figure 14b) has the same effect, and the model with the smallest RMSE in Hunan province (Figure 14c) also has four categories, while the combination of the summer model *R*^2^ shows that the GWRK model that considers ZTD has the best effect overall. In Guangdong province (Figure 14d), the RMSEs of various models were not very different, but combined with the magnitudes of the *R*^2^ values, it can be judged that the GWR-EBK model considering ZTD is the next-best model, and the GWR-EBK model considering temperature and ZTD is the best model in Guangxi province (Figure 14e).

The two types of GWR-EBK models considering temperature and ZTD had the best accuracy performances in Henan (Figure 14c) and Hubei provinces (Figure 14b) in autumn, while the GWR-EBK model considering ZTD had the best effect in Hunan province (Figure 14c), the GWR-EBK model constructed by combining three variables had the best and equal effect in Guangdong province (Figure 14d), and the two types of GWR-EBK models considering PWV and temperature had the best effects in Guangxi province (Figure 14e).

## 6. Discussion

After comparing the interpolation effects of three interpolation methods in obtaining meteorological data and GNSS-derived PWV and ZTD data from PM_2.5_ ground stations, the results show that the IDW interpolation method has better interpolation effects than the kriging method and the TSF interpolation method for meteorological parameters and GNSS-derived PWV and ZTD data in the five studied central and southern provinces.

By comparing the changes in the spatial distribution map of PM_2.5_ in the region, it can be seen that the PM_2.5_ concentration values in the five south-central provinces of China show obvious seasonal variation characteristics of high values in winter and low values in summer, and geographical characteristics of high values in the north and low values in the south, while clustering phenomena mainly occur in Henan and Guangdong provinces, with high clustering in Henan province and low clustering in Guangdong province recorded.

The experiments show that considering GNSS-derived PWV and ZTD data in the construction of a regional PM_2.5_ model is effective and feasible and is better than the regional model constructed by considering only the atmospheric pollutants, meteorological factors and elevation factors in the annual and seasonal averages in many cases, with better interpolation effects. In the five provinces of south-central China, PWV and ZTD show strong negative correlations with PM_2.5_ as well as with temperature, showing seasonal characteristics of low values in winter and high values in summer, and geographical characteristics of high values in the south and low values in the north. The significance magnitude in the exploratory regression is divided into atmospheric pollutants > GNSS parameters (ZTD and PWV) > meteorological factors, and the results of the stratified heterogeneity of geodetectors again reflected the significant correlation between GNSS parameters and PM_2.5_, providing a guide for constructing a regional PM_2.5_ model when meteorological data are missing and PWV and ZTD data can be used as a substitute.

In constructing a regional model of PM_2.5_, the interpolation effect of the model changes depending on the choice of variables, the time scale, and the spatial scale. The GWR model has a stronger ability to estimate PM_2.5_ than the MGWR model and is more efficient in south-central China. Compared with the interpolation effect of a single geographically weighted regression-type model for PM_2.5_, the combined model shows a stronger advantage, and the overall best performance in this area is obtained with the GWR-EBK model, indicating that the empirical Bayesian kriging method is better for the explanation and interpretation of GWR residuals; further, the GWR-EBK model can improve the accuracy by 14.74% more than the GWR model.

The largest reason for the inverse ratio of the seasonal PM_2.5_ values to the seasonal average scale is that when the PM_2.5_ values are lower due to seasonal changes, the reduction ratio of the sum of squares of the difference between the true value and the mean value may be greater than the reduction ratio of the sum of squares of the residuals, as shown by the formula used to calculate the *R*^2^ values. This explains why the *R*^2^ value obtained in winter is greater than those obtained in spring and autumn, while the *R*^2^ value in summer is the smallest.

## 7. Conclusions

This article analyzed the spatial and temporal characteristics of PM_2.5_ in the five provinces of south-central China in 2017–2019 on two time scales (annual average and seasonal average); introduced two types of variables, GNSS-derived PWV and ZTD variables, to participate in the construction of the regional model of PM_2.5_ in the region; analyzed a total of six types of models constructed by the combination of three variables, and obtained the following conclusions:

(1) The IDW interpolation method is most suitable for the regional interpolation of meteorological parameters and GNSS-derived PWV and ZTD data for the five studied provinces in south-central China.

(2) The PM_2.5_ concentration values in the five south-central provinces show clear characteristics of high values in the north and low values in the south, as well as a seasonal variation pattern of high values in winter and low values in summer; clustering phenomena mainly occur in Henan and Guangdong provinces, with high clustering in Henan province and low clustering in Guangdong province recorded.

(3) The GNSS-derived PWV and ZTD data show a strong negative correlation with PM_2.5_ in the five provinces in south-central China, so it is effective and feasible to consider GNSS-derived PWV and ZTD in the construction of a regional model of PM_2.5_, and the experiments show that the interpolation is better than the regional model constructed by considering only atmospheric pollutants and meteorological and elevation factors with many iterations at both the annual and seasonal average scales.

(4) Compared with a single geographically weighted regression-type model for PM_2.5_, the combined model shows a stronger advantage, and the best overall performance in the five south-central provinces of China is obtained with the GWR-EBK model, indicating that the empirical Bayesian kriging method has a stronger ability to explain the GWR residuals and a better interpolation effect.

(5) In the five provinces of south-central China, the applicability of the model in higher-elevation areas is not as good as that in lower-elevation areas, and the applicability of the model in higher-latitude areas is also worse than in lower-latitude areas.

## Figures and Tables

**Figure 1 ijerph-18-07931-f001:**
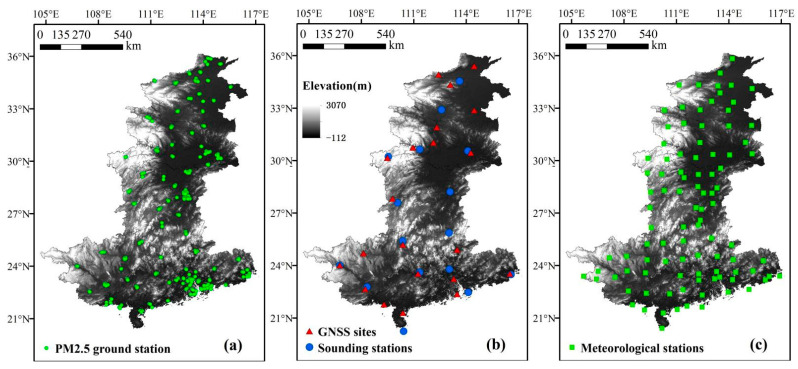
Distribution of observation stations. GNSS: global navigation satellite system. (**a**) PM_2.5_ ground station, (**b**) GNSS sites and Sounding stations, (**c**) Meteorological stations.

**Figure 2 ijerph-18-07931-f002:**
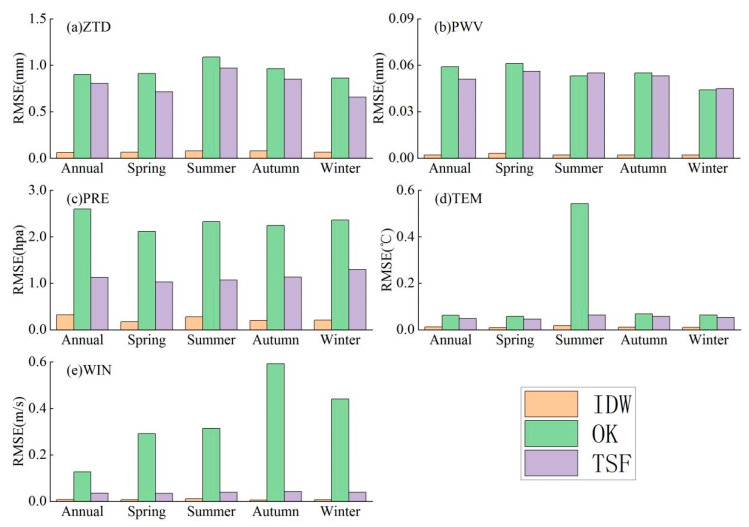
Interpolation effect comparison: (**a**–**e**) Results of root mean square error of three interpolation methods (IDW, OK, TSF) for interpolating five variables (ZTD, PWV, PRE, TEM and WIN) respectively. Note: DEM means the elevation values obtained from the digital elevation model, WIN means wind, PRE means pressure, TEM means temperature, ZTD means zenith tropospheric delay, PWV means precipitable water vapor, IDW means inverse distance weighting, OK means ordinary kriging, TSF means the tension spline function.

**Figure 3 ijerph-18-07931-f003:**
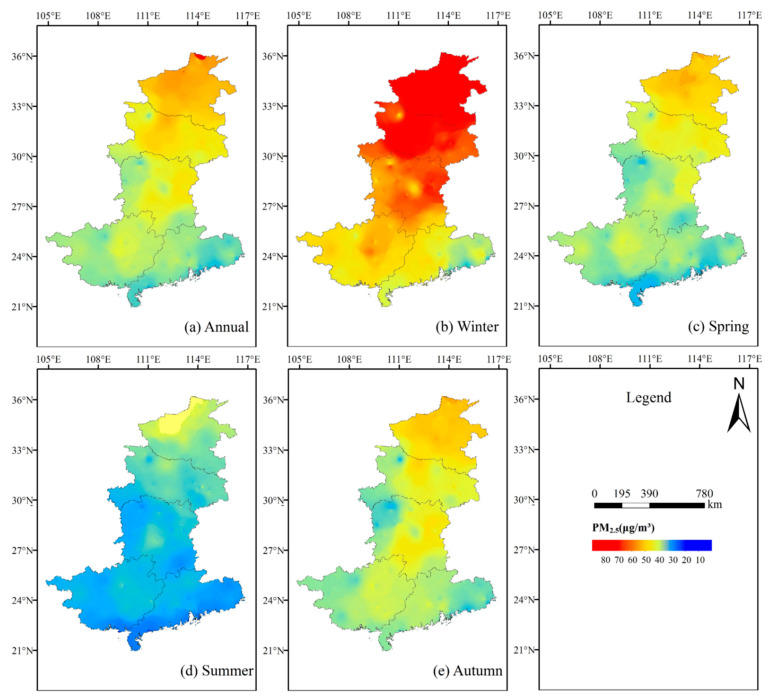
Distribution of PM_2.5_: (**a**) Annual, (**b**) Winter, (**c**) Spring, (**d**) Summer, (**e**) Autumn.

**Figure 4 ijerph-18-07931-f004:**
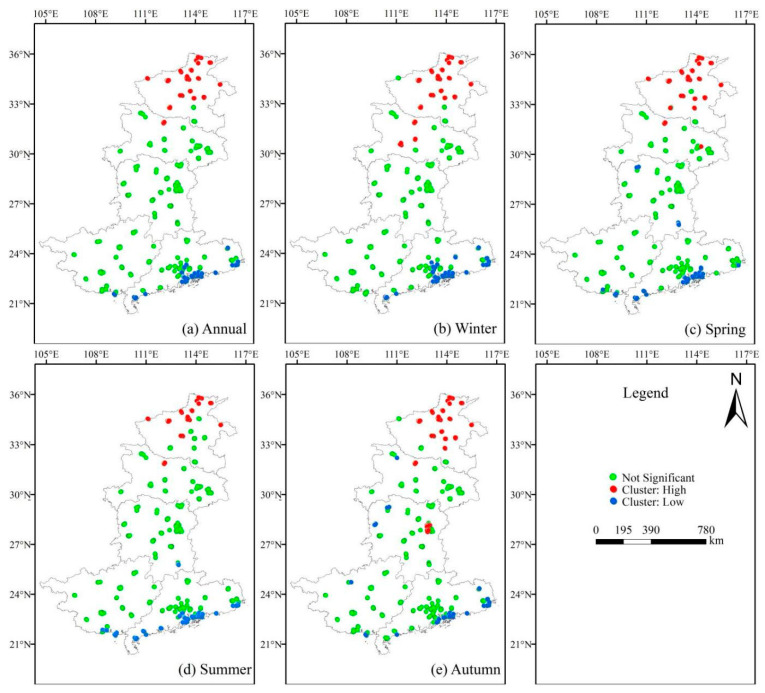
Clusters of PM_2.5_: (**a**) Annual, (**b**) Winter, (**c**) Spring, (**d**) Summer, (**e**) Autumn.

**Figure 5 ijerph-18-07931-f005:**
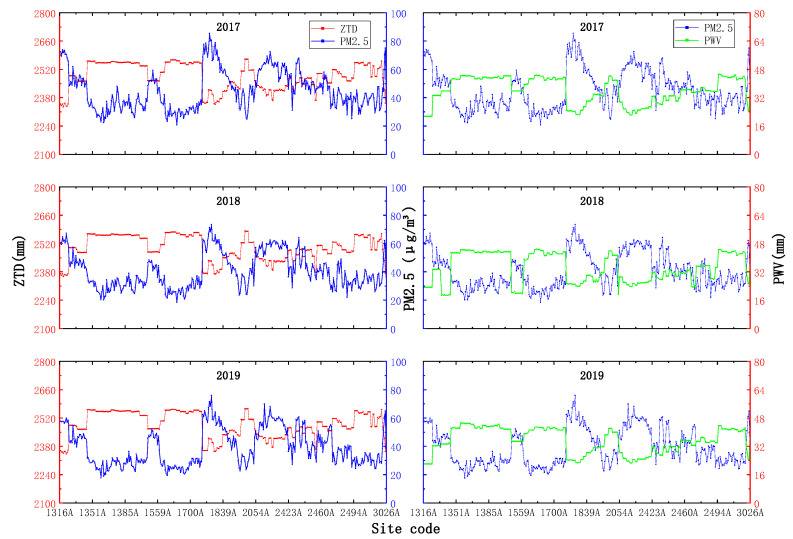
Annual average scale ZTD, PWV and PM_2.5_ changes.

**Figure 6 ijerph-18-07931-f006:**
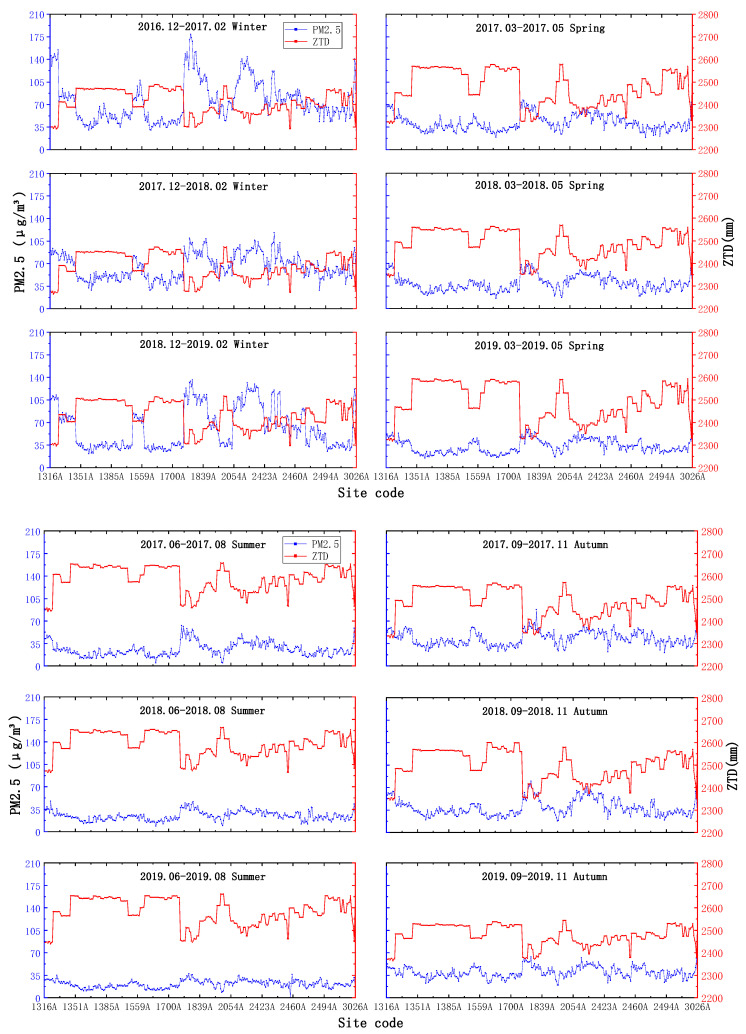
Seasonal average-scale ZTD and PM_2.5_ changes.

**Figure 7 ijerph-18-07931-f007:**
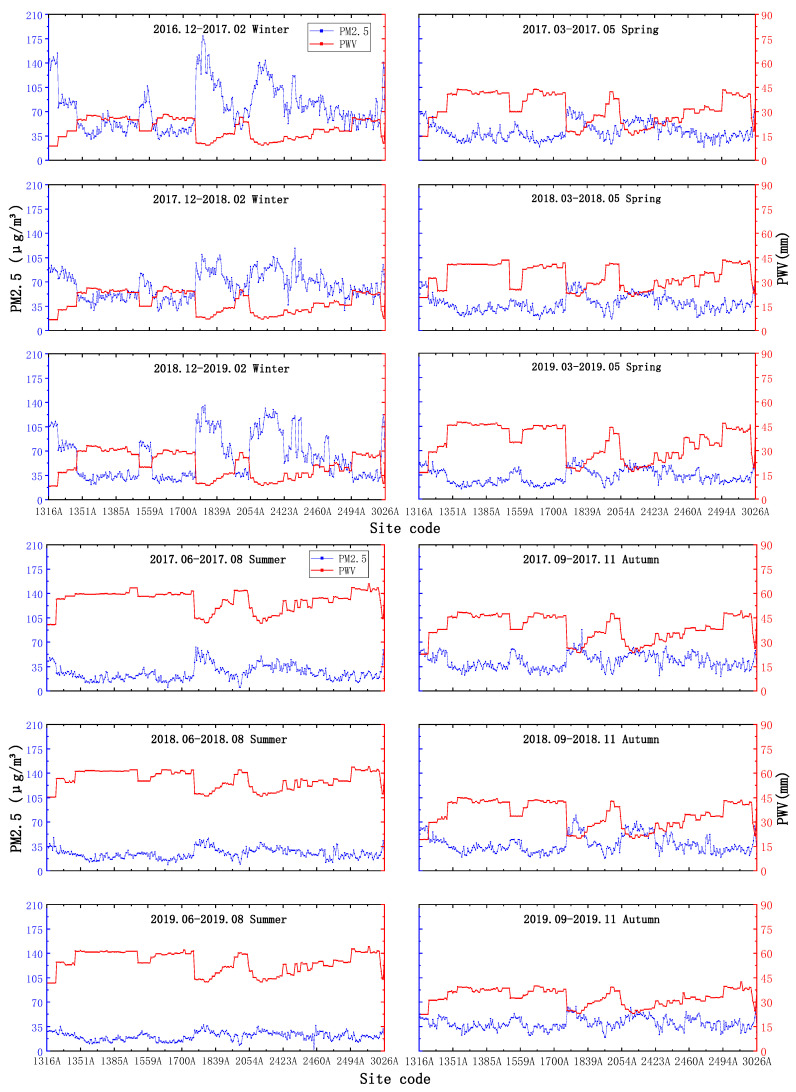
Seasonal average-scale PWV and PM_2.5_ changes.

**Figure 8 ijerph-18-07931-f008:**
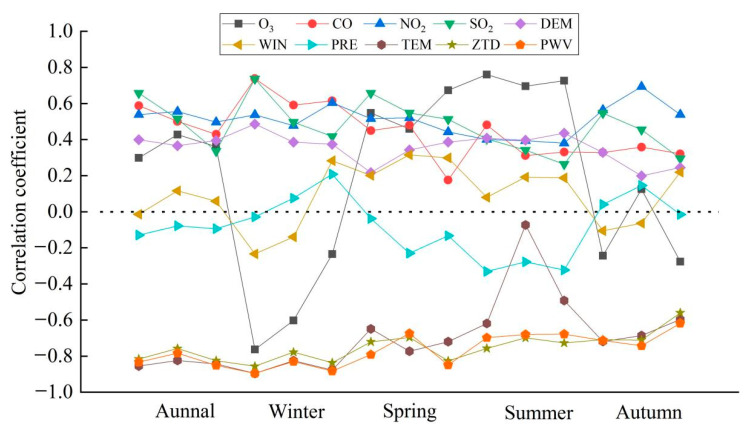
Spearman’s correlation coefficient.

**Figure 9 ijerph-18-07931-f009:**
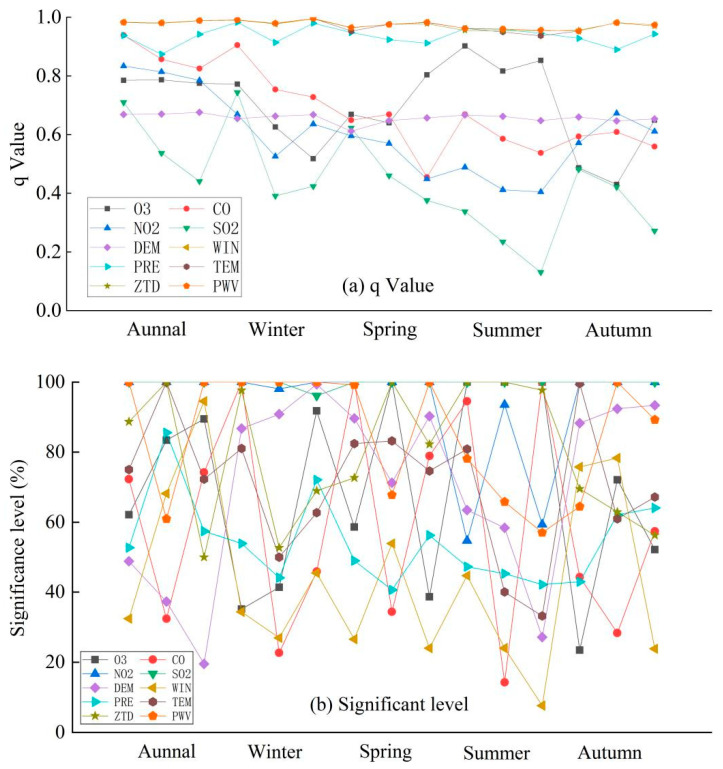
Variable correlation degrees: (**a**) *q* values, (**b**) significant levels.

**Figure 10 ijerph-18-07931-f010:**
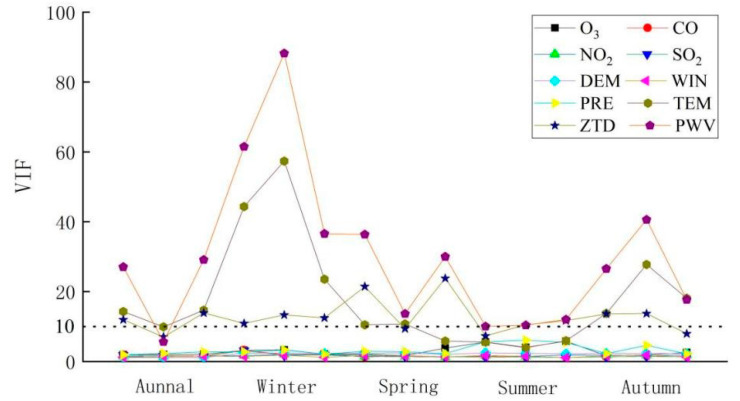
The results of the multicollinearity test. Magnitude of VIF (variance inflation factor) values for O_3_, CO, NO_2_, SO_2_, DEM, WIN, PRE, TEM, ZTD and PWV at different time stamps. Note: DEM means the elevation values obtained from the digital elevation model, WIN means wind, PRE means pressure, TEM means temperature, ZTD means zenith tropospheric delay, PWV means precipitable water vapor.

**Figure 11 ijerph-18-07931-f011:**
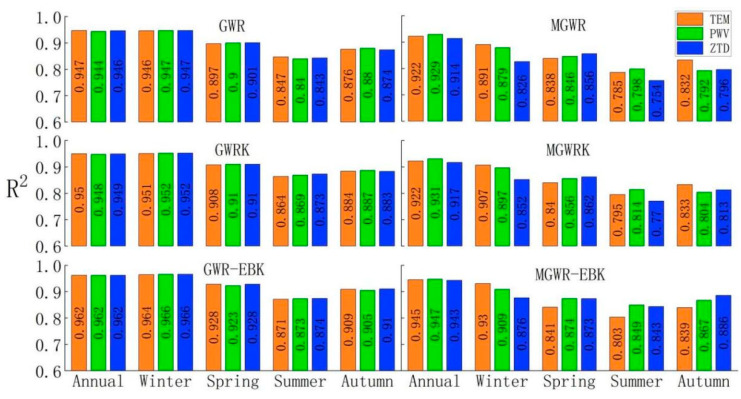
Models’ *R*^2^ values.

**Figure 12 ijerph-18-07931-f012:**
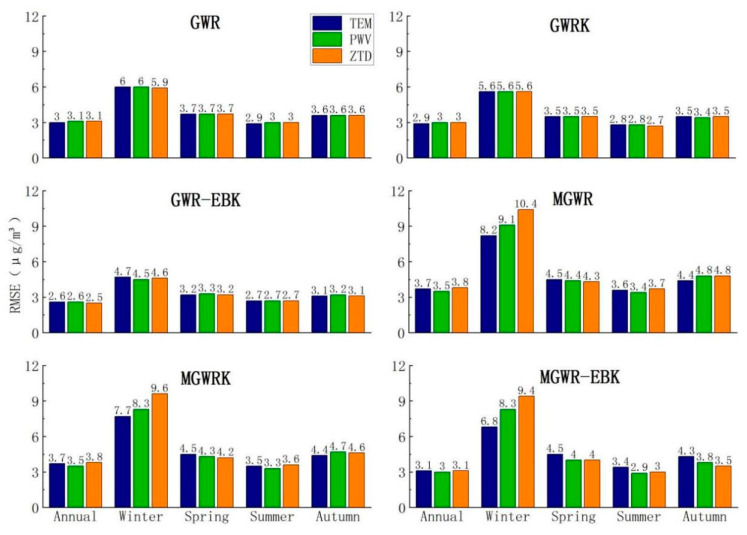
The RMSE values of the overall results obtained by the interpolation for the six models.

**Figure 13 ijerph-18-07931-f013:**
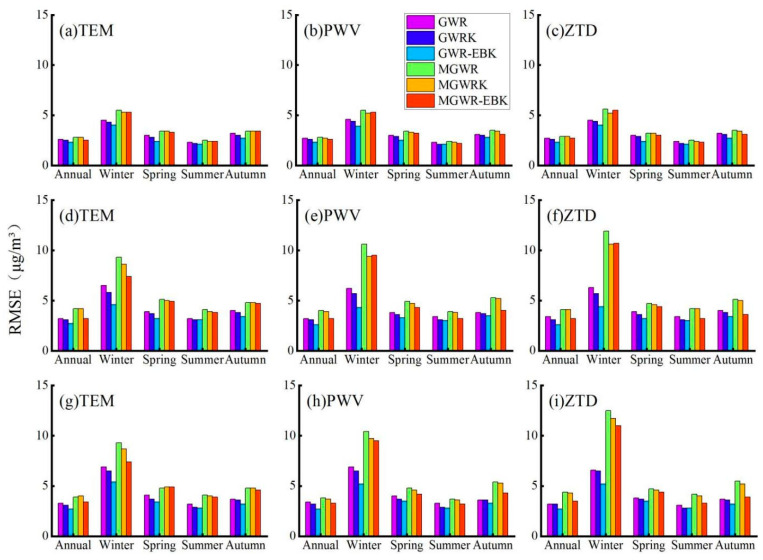
Model RMSEs for different elevation intervals: (**a**–**c**) ≤35 m, (**d**–**f**) 35–90 m, (**g**–**i**) ≥90 m.

**Figure 14 ijerph-18-07931-f014:**
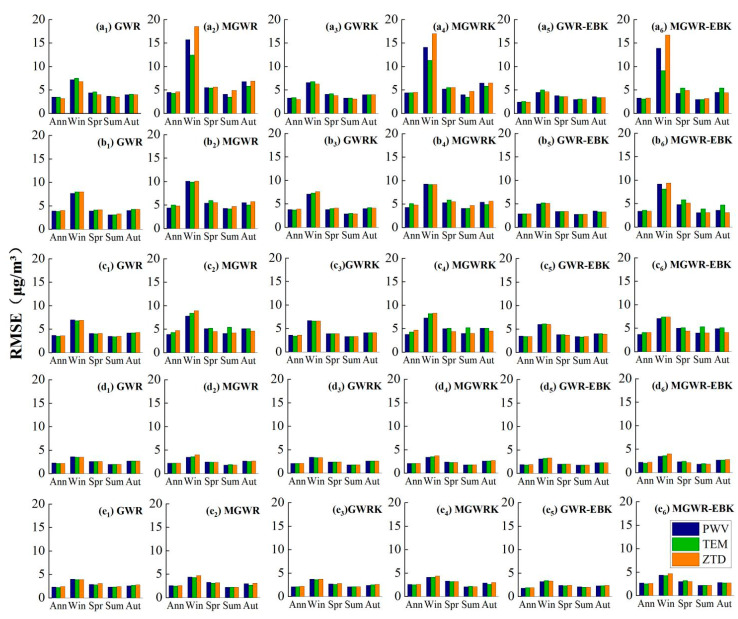
Model RMSEs for different provinces: (**a**) Henan, (**b**) Hubei, (**c**) Hunan, (**d**) Guangdong, (**e**) Guangxi.

**Table 1 ijerph-18-07931-t001:** Site Data Introduction.

Type	Time	O_3_	CO	NO_2_	SO_2_	DEM	WIN	PRE	TEM	ZTD	PWV
Annual	2017	340 PM_2.5_ ground-based stations					97 meteorological monitoring stations			21 GNSS sites	16 sounding stations
	2018										
	2019										
Winter	December 2016–February 2017										
	December 2017–February 2018										
	December 2018–February 2019										
Spring	March 2017–May 2017										
	March 2018–May 2018										
	March 2019–May 2019										
Summer	June 2017–August 2017										
	June 2018–August 2018										
	June 2019–August 2019										
Autumn	September 2017–November 2017										
	September 2018–November 2018										
	September 2019–November 2019										

Note: DEM means the elevation values obtained from the digital elevation model, WIN means wind, PRE means pressure, TEM means temperature, ZTD means zenith tropospheric delay, PWV means precipitable water vapor, GNSS means global navigation satellite system.

**Table 2 ijerph-18-07931-t002:** Variable Combination Scheme.

	Combination							
One	O_3_	CO	NO_2_	SO_2_	DEM	WIN	PRE	TEM
Two	O_3_	CO	NO_2_	SO_2_	DEM	WIN	PRE	PWV
Three	O_3_	CO	NO_2_	SO_2_	DEM	WIN	PRE	ZTD

Note: DEM means the elevation values obtained from the digital elevation model, WIN means wind, PRE means pressure, TEM means temperature, ZTD means zenith tropospheric delay, PWV means precipitable water vapor.

## Data Availability

Not applicable.

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
