# Peer review of "Ingestion of GNSS-Derived ZTD and PWV for Spatial Interpolation of PM_2.5_ Concentration in Central and Southern China"

_ijerph, 2021, doi:10.3390/ijerph18157931_

Round 1
Reviewer 1 Report
I enjoyed reading through the novel process of data analysis presented in this manuscript. The mathematical rigor of presented here is of good quality. The presentation of the overall content can be improved further for clarity in the message being delivered.
Major comments:
- The abstract can be clearer about where the PM5 data is actually coming from. Not in an elaborate form as mentioned between lines 272-283, but at least mentioning that they came from ground-based stations would be helpful.
- More clarity is expected for the direct message to be delivered across.
- Please closely follow journal formatting guidelines, maintain line-spacing between the body of text and figures, pay attention to the excessive indentation etc. (I suggest getting the manuscript reviewed by a technical editor first).
Minor/technical/editorial comments:
- I suggest getting rid of the numbers in parentheses (i.e., (1), (2), (3), (4), …) in the abstract
- Please maintain consistency in the usage of subscripts in PM5
- Acronyms need to be spelled out on their first use. Several acronyms are either never defined or defined somewhere in the middle of the text where it has already been referred to early on several times (e.g., ZWD).
- Please maintain consistency of the font sizes across all figures.
- Some figures (e.g., Figure 9) has some data points cropped out partially. Please include all data points within figures.

Author Response
Dear reviewer:
Thank you for your letter and for the reviewers’comments concerning our manuscript entitled“Paper Title”(ID:ijerph-1311791).Those comments are all valuable and very helpful for revising and improving our paper, as well as the important guiding significance to our researches. We have studied comments carefully and have made correction which we hope meet with approval.Revised portion are marked in red in the paper.
I've put the response to your revision in the attachment for your review, best wishes to you!

Reviewer 2 Report
Dear authors,
see the attached file for comments and suggestions.
Keep the good work!

Author Response
Dear Reviewer:
Thank you for your letter and for the reviewers’comments concerning our manuscript entitled“Paper Title”(ID:ijerph-1311791).Those comments are all valuable and very helpful for revising and improving our paper, as well as the important guiding significance to our researches. We have studied comments carefully and have made correction which we hope meet with approval.Revised portion are marked in red in the paper.
I've put the response to your revision in the attachment for your review, best wishes to you!
